# Systemic Expression of Genes Involved in the Plant Defense Response Induced by Wounding in *Senna tora*

**DOI:** 10.3390/ijms221810073

**Published:** 2021-09-17

**Authors:** Ji-Nam Kang, Woo-Haeng Lee, So Youn Won, Saemin Chang, Jong-Pil Hong, Tae-Jin Oh, Si Myung Lee, Sang-Ho Kang

**Affiliations:** 1Genomics Division, National Institute of Agricultural Sciences, Rural Development Administration, Jeonju 54874, Korea; greatnami@korea.kr (J.-N.K.); soyounwon@korea.kr (S.Y.W.); schang44@korea.kr (S.C.); hjp3467@korea.kr (J.-P.H.); 2Department of Life Science and Biochemical Engineering, Sun Moon University, Asan 31460, Korea; kain832@naver.com (W.-H.L.); tjoh3782@sunmoon.ac.kr (T.-J.O.)

**Keywords:** wounding, systemic defense response, jasmonate biosynthesis, ethylene biosynthesis, transcription factors, pathogenesis related genes, flavonoid biosynthesis, *Senna tora*

## Abstract

Wounds in tissues provide a pathway of entry for pathogenic fungi and bacteria in plants. Plants respond to wounding by regulating the expression of genes involved in their defense mechanisms. To analyze this response, we investigated the defense-related genes induced by wounding in the leaves of *Senna tora* using RNA sequencing. The genes involved in jasmonate and ethylene biosynthesis were strongly induced by wounding, as were a large number of genes encoding transcription factors such as ERFs, WRKYs, MYBs, bHLHs, and NACs. Wounding induced the expression of genes encoding pathogenesis-related (PR) proteins, such as PR-1, chitinase, thaumatin-like protein, cysteine proteinase inhibitor, PR-10, and plant defensin. Furthermore, wounding led to the induction of genes involved in flavonoid biosynthesis and the accumulation of kaempferol and quercetin in *S*. *tora* leaves. All these genes were expressed systemically in leaves distant from the wound site. These results demonstrate that mechanical wounding can lead to a systemic defense response in the Caesalpinioideae, a subfamily of the Leguminosae. In addition, a co-expression analysis of genes induced by wounding provides important information about the interactions between genes involved in plant defense responses.

## 1. Introduction

Wounding is a common type of damage in plants caused by abiotic stresses, such as wind, rain, and hail, or by the feeding of insects and other herbivores. Tissues destroyed by wounding provide a pathway for pathogen infection, which threatens plant survival [1,2]. To mitigate this problem, plants have evolved defense mechanisms that effectively cope with wound damage [1]. Plant defense mechanisms against wounding and fungal attacks share many genes in common, including those involved in hormone biosynthesis, pathogen defense, transcriptional regulation, and secondary metabolic pathways [1,2]. The wound response comprises two mechanisms in plants: the first is a local response in the damaged tissue, and the second is a systemic response in undamaged tissues [3,4]. The latter requires the transmission of a wound signal. Systemin, comprising 18 amino acids, is a peptide that accumulates in cells following wounding or insect attack, and which has been recognized as a molecule for long-distance wound signaling [1,3,4]; however, this is not the case for all plant species. Studies in tomato (*Solanum lycopersicum*), for example, have shown that systemin is not a long-distance systemic transmission signal in this species [4,5]. Many studies have suggested that jasmonates are one part of the long-distance transport of the wound signal [3,4,6].

Jasmonates are typical wound-inducible hormones, which can induce defense mechanisms in plants [1,2,6,7]. Jasmonates, such as jasmonic acid (JA), isoleucine conjugate JA (JA-Ile), and methyl ester JA (MeJA), rapidly and transiently accumulate in the wound site and also act as signaling molecules for the systemic defense response [3,4,5,6]. The jasmonate-mediated response requires the production of fatty acids, a substrate for the octadecanoid pathway [3,8]. The main enzymes involved in the jasmonate biosynthesis pathway are well known. The first step of jasmonate biosynthesis occurs in the plastid, where the sequential activities of lipoxygenase (LOX), allene oxide synthase (AOS), and allene oxide cyclase (AOC) convert linolenic acid to oxo-phytodienoic acid (OPDA). OPDA is then converted to 3-oxo-2 (2′[Z]-pentenyl)-cyclopentane-1-octanoic acid (OPC8) by OPDA reductase (OPR3), then into OPC8-CoA by 4-coumarate-CoA ligase (OPCL1). Finally, JA is produced through beta-oxidation by acyl-coenzyme A oxidase (ACX), fatty acid beta-oxidation multifunctional protein (AIM), and 3-ketoacyl-CoA thiolase (KAT) [5,7,8,9,10]. Furthermore, JA-Ile, the main jasmonate component of defense responses, derives from JA by jasmonate resistant 1 (JAR1) [7,11]. MeJA, another active form of jasmonate, is produced by JA carboxyl methyltransferase (JMT) [5]. Jasmonates regulate many aspects of plant defense responses [12], including reprograming the expression of defense-related genes [13].

Many studies have reported that wounding activates many defense-related genes, such as those encoding transcription factors (TFs), pathogenesis-related (*PR*) genes, and flavonoid biosynthesis genes. ERFs, WRKYs, MYBs, NACs, and bHLHs are well known TFs in plants [14,15,16], all of which are controlled by wounding to function as important transcriptional regulators of defense-related genes in plants [2,15,16,17]. Many *PR* genes can be triggered by wounding [1,2,18]. PR-1, chitinase (CHT), thaumatin-like protein (TLP), proteinase inhibitor (PI), plant defensin (PDF), and PR-10 are PR proteins [18]. Flavonoids are essential secondary metabolites in plants, which are synthesized via the phenylpropanoid biosynthesis pathway. Flavonoids are involved in protection against biotic and abiotic stress [13,19,20]. Wounding leads to the production of flavonoids [21,22]. Phenylalanine ammonia-lyase (PAL), cinnamate 4-hydroxylase (C4H), 4-coumaroyl-CoA ligase (4CL), chalcone synthase (CHS), chalcone isomerase (CHI), flavonone 3-hydroxylase (F3H), flavonoid 3′-monooxygenease (F3′H), and flavonol synthase (FLS) are major enzyme components of the flavonoid biosynthesis pathway [23,24,25].

*Senna tora* is an annual shrub belonging to the Caesalpinioideae, a subfamily of the Leguminosae, and is mainly used as medicinal food source in India, China, Sri Lanka, Nepal, and Korea [26]. However, our understanding of the defense mechanisms of this species is limited. Recently, high-quality *S*. *tora* genome sequences were published [27]. In this study, we comprehensively analyzed wound-induced defense-related genes in the leaves of this plant using RNA sequencing (RNA-seq). These results demonstrate that mechanical wounding induces the systemic expression of many defense-related genes in the Caesalpinioideae. This study provides insights into the complex defense mechanisms regulated by wounding in plants.

## 2. Results and Discussion

### 2.1. Identification of Differentially Expressed Genes (DEGs) following Wounding

RNA-seq was performed on *S*. *tora* leaves collected at 1 (T1), 3 (T3), 6 (T6), and 24 (T24) hours after wounding, as well as control leaves (T0). After preprocessing the raw reads, 28,255,742–44,084,054 clean reads were obtained from each sample. The overall GC ratio was 45.1% on average, and the Q30 values were 96.05–96.73%. The reads were then mapped onto the *S*. *tora* reference genome [27], with an average mapping ratio of 97.8% (Appendix A).

Analysis of differentially expressed genes (DEGs) was performed based on fragments per kilobase of transcript per million mapped reads (FPKM) values. Genes showing an expression fold change (FC) >2 relative to T0 were considered DEGs at each timepoint (Figure 1A). A total of 2073, 5060, 4976, and 1682 genes were identified as DEGs at the T1, T3, T6, and T24 timepoints after wounding, respectively. The highest number of DEGs was found in T3, with a similar number identified in T6 (Figure 1B). A total of 453 genes (12.39%) were continuously upregulated between T1 and T6, while 716 genes (19.59%) were positively induced from T3 to T6. Among the DEGs identified in T1, only 33 genes (0.9%) maintained a positive upregulation at T24 after wounding (Figure 1C). Similar results were also found in the downregulated DEGs (Figure 1D).

Analyses of Gene Ontology (GO) term enrichment and Kyoto Encyclopedia of Genes and Genomes (KEGG) term enrichment were performed on the DEGs to predict their functions. The DEGs were grouped into the three main GO categories, consisting of biological processes, cellular components, and molecular functions. Most of the DEGs were associated with GO terms such as metabolic process, cellular process, biological regulation, membrane and cell parts, catalytic activity, and binding activity. These GO terms were most enriched between T3 and T6 after wounding (Figure 2A). A detailed GO enrichment analysis was performed on the upregulated DEGs associated with the biological process term. Genes associated with the regulation of JA and salicylic acid (SA) biosynthesis, the bacterium response, and phytoalexin biosynthesis were activated in T1. Fungal and bacterial responses and the chorismate metabolic process were the dominant GO terms between T1 and T6 after wounding; however, these GO terms were not observed at T24. Negative regulators of peptidase, phosphatase, and hydrolase activities were the main GO terms observed at T24 after wounding (Figure 2B). KEGG analysis indicated that the main pathways activated by wounding were alpha-linolenic acid metabolism and the flavonoid biosynthesis pathway. The alpha-linolenic acid metabolic pathway was activated in T1 and was maintained up to T3. Flavonoid biosynthesis was induced in T1 and remained active until T6. These pathways were not still activated at T24 after wounding (Figure 2D). These results indicate that the defense responses induced by wounding are mainly activated within T6 in *S*. *tora* leaves. Comprehensive information on wound-responsive genes is limited in legumes; however, in Arabidopsis (*Arabidopsis thaliana*), wounding induces many defense-related genes, including those encoding various TFs; PR proteins; and members of the phenylpropanoid biosynthesis, jasmonate, and ethylene (ET) pathways within 6 h [2]. 

On the other hand, the GO and KEGG terms involved in the photosynthesis process were negatively modulated after wounding (Figure 2C,E). Reduction of photosynthetic activity by biotic and abiotic stresses has been reported [28,29]. In addition, the indoleacetic acid metobolism and borate transport were the major GO terms inhibited by wounding in *S. tora* leaves (Figure 2C). These biological processes are closely related to plant growth. Indolacetic acid is a major auxin in higher plants, which is important for plant growth and development [30]. Borate is an important micronutrient for plant growth [31]. These results suggest that the wound response negatively regulates the biological processes involved in plant growth. It is believed that wounding can lead to allocating resources from growth to defense in plants [29].

### 2.2. Expression Analysis of Genes Involved in Plant Hormone Biosynthesis following Wounding

When plant tissue is damaged, wound-induced responses can be triggered by the de novo biosynthesis of jasmonates [11,32]. Since jasmonates are recognized as typical wound-inducible hormones, we investigated all genes involved in jamonate biosynthesis and metabolism in *S*. *tora* leaves. Twenty-one of these genes were identified as DEGs after wounding. Among them, genes encoding LOX3 (*STO02G025740*), AOS1 (*STO09G315120*), AOC3 (*STO10G322150*), OPR3 (*STO03G063940*), OPCL1 (*STO05G118310*), ACX1 (*STO11G368240*), AIM1 (*STO08G252570*), KAT2 (*STO01G024280*), JAR1 (*STO06G173150*), JMT (*STO06G200560* and *STO11G364450*), CYP94B (*STO12G393970*), and CYP94C1 (*STO07G213990* and *STO12G378540*) were strongly induced in T1 (Figure 3A,B and Appendix A). Interestingly, these genes were systemically induced in undamaged leaves distant from the initial wound site. The additional wound treatment between T3 and T24 no longer elicited strong expression of jasmonate biosynthesis genes (Figure 3B and Appendix A), suggesting that the first wound treatment triggered the de novo expression of genes involved in jasmonate biosynthesis in the undamaged leaves (T1). Koo et al. (2009) demonstrated that the systemic accumulation of jasmonates in undamaged tissues did not involve their transportation from the wound site [11]. Jasmonates are believed to be biosynthesized following systemin recognition around the wounding site, after which the jasmonates are transported to the undamaged tissues. This jasmonate signal can further lead to de novo jasmonate biosynthesis in the undamaged tissues, thereby activating the systemic expression of jasmonate-responsive defense genes [1,4,5,6,11].

Furthermore, the genes responsible for MeJA production were significantly upregulated, while JA-Ile biosynthesis was inhibited. In T1, we observed the strong induction of two JMT-encoding genes, *STO06G200560* and *STO11G364450*, with FC values approximately 143 times and 7.1 times that of T0, respectively. Expression of the JAR1-encoding gene *STO06G173150* was approximately 1.7 times higher in T1 than in T0. Along with the relatively weak induction of the JAR1-encoding gene, strong induction of *STO12G393970*, *STO12G378540*, and *STO07G213990* expression was identified in T1, leading to the production of CYP94B3 and CYP94C1 (Figure 3B and Appendix A). These genes have been reported to be involved in the oxidation of JA-Ile [33,34].

Furthermore, genes encoding the TIFY family were strongly induced in T1 (Figure 3C and Appendix A). TIFY proteins are jasmonate ZIM domain-containing (JAZ) proteins, which transcriptionally repress JA-Ile-responsive genes in the absence of JA-Ile [5,12,35]. JA-Ile mediates the interaction between coronatine insensitive 1 (COI1) and JAZ proteins, thereby causing the degradation of the JAZ protein by the 26S proteasome. The degradation of the JAZ protein can ultimately activate the transcription of JA-Ile-responsive genes [5,7]. Of all the jasmonates, only JA-Ile has been reported to mediate the interaction of JAZ proteins and COI1 [11]. The high expression of the TIFY-encoding genes and the genes encoding CYP94B3 and CYP94C1 are both inhibitory mechanisms of JA-Ile activity [12,35]. These results suggest that MeJA may be more important than JA-Ile for the expression of jasmonate-responsive genes in *S*. *tora* leaves.

MeJA can drive the expression of many defense-related genes in plants [7,13,16]. It is also highly volatile, and can easily penetrate cell membranes, which is advantageous for spreading as a signal to distant leaves [5]. Chung et al. (2008) suggested that JA derivatives other than JA-Ile may modulate jasmonate-mediated plant defense responses [12]. In tomato, the *jar1* mutant only produces about 20% of the JA-Ile of the wild type; however, its expression of the wound-response genes is similar to that of the wild type. These authors suggested that other wound-inducible jasmonates therefore activate the expression of the wound-response genes in the *jar1* mutant [11]. In addition, low JA-Ile activity may induce the defense responses following wounding. Since a failure to regulate JA-Ile causes serious defects in plant growth, it is essential to control the massive amount of JA-Ile produced by wounding [35]. Induction of the JAZ proteins can prevent the hyperactivity of JA-Ile [35,36]. Koo et al. (2009) showed that the remaining JA-Ile activity in the tomato *jar1* mutant can enable the sufficient expression of wound-responsive genes [11].

Additionally, we examined the expression of genes involved in the ET, SA, and abscisic acid (ABA) pathways (Figure 4A and Appendix A). ET production is induced by wounding, and is crucial in plant disease resistance [1,37,38,39,40]. The ET biosynthesis process is enzymatically simple in plants: ET is produced from S-adenosyl-methionine by two enzymes, 1-aminocyclopropane-1-carboxylate synthase (ACS) and 1-aminocyclopropane-1-carboxylate oxidase (ACO) [39]. A significant induction of the ET biosynthesis genes was observed between T1 and T24 after wounding compared with their expression at T0. The expression patterns of the paralog genes differed temporally. Expression of the ACS-encoding gene *STO06G193910* and two ACO-encoding genes, *STO09G290810* and *STO09G290820*, was induced in T1 and peaked at T3. On the other hand, the expression of their paralog genes, *STO01G012020* and *STO11G353060*, was induced in T3, and their significant expression was maintained until T24 compared with that of T0 (Figure 4B,C). These results are consistent with the fact that wounding enhanced the expression of the *ACO* genes in mung bean (*Vigna radiata*) [41] and indicated that ET production by wounding is temporally regulated in *S*. *tora* leaves. ET can enhance the systemic wound response via the octadecanoid pathway [3]. The relatively early induction of ET biosynthesis is expected to alter the expression of regulatory proteins, such as TFs [2]. ET is also required for the expression of *PR* genes [40], many of which encode enzymes activated relatively late in the plant defense responses [2]. Relatively late-biosynthesized ET is therefore required for the production of the PR proteins.

SA levels increase during infection with viruses, fungi, and bacteria, as well as herbivory by insects, which is essential for the systemic acquired resistance response [38,42]. Two SA biosynthesis pathways, isochorismate and phenylalanine, have been reported in plants, which are both derived from chorismate [42]. The first pathway involves isochorismate synthase (ICS), PphB SUSCEPTIBLE (PBS), and ENHANCED PSEUDOMONAS SUSCEPTIBILITY (EPS), which is the main pathway for SA biosynthesis in Arabidopsis [42]. The second is the pathway involving chorismate mutase (CM), PAL, and AIM, which has been identified as a major pathway for SA biosynthesis in rice (*Oryza sativa*) [42]. In the present study, the genes involved in the first SA pathway were not upregulated by wounding (Appendix A); however, the genes involved in the second SA pathway were strongly induced by wounding. Two CM-encoding genes, *STO09G312730* and *STO08G259570*, the PAL1-encoding gene *STO06G182100*, and the AIM-encoding gene *STO08G252570* showed similar expression patterns among the SA biosynthesis genes (Figure 4B,D). Considering that the phenylalanine production from chorismate branches into many other biosynthesis pathways, PAL is considered an upstream enzyme of another defense-related biosynthesis pathway, AIM is involved in the beta-oxidation of jasmonates during their biosynthesis [8,42,43], so the expression of these genes may not be altered to promote SA biosynthesis but for other reasons. Nevertheless, it is interesting that the strong expression of these genes was induced in T3 after wounding (Figure 4D). At this timepoint, the expression of genes involved in jasmonate biosynthesis was remarkably reduced compared with that of T1 (Figure 3B and Appendix A). The biosynthesis of jasmonates and SA generally negatively regulate each other [3]: SA biosynthesis is likely to occur when jasmonate biosynthesis is limited in *S*. *tora* leaves.

ABA is produced from its precursor zeaxanthin by the sequential enzymatic activities of zeaxanthin epoxidase (ZEP), 9-cis-epoxycarotenoid dioxygenase (NCED), short-chain dehydrogenase/reductase (SDR), and ABA aldehyde oxidase (AAO) in plants [22,44]. The genes involved in ABA biosynthesis were not positively stimulated by wounding in the *S*. *tora* leaves study (Appendix A). ABA is not considered a primary signal in the wound response [3].

Plant defense mechanisms are mediated by a complex hormone network in plants [37,38]. Jasmonates and ET are required for the adaptation of plants to environmental stresses [3,5,40], and are expected to be the main hormones induced by wounding in *S*. *tora* leaves. SA may also be involved in the defense mechanism following wounding in this species.

### 2.3. Induction of Genes Encoding Transcription Factors following Wounding

Many genes encoding ERF, MYB, WRKY, bHLH, and NAC TFs were systemically induced by wounding in *S*. *tora* leaves (Figure 5 and Appendix A). Forty-one genes encoding WRKY TFs were upregulated following wounding, which was the largest group of the wound-inducible TFs (Figure 5A and Appendix A). In Arabidopsis, several *WRKY* genes were induced by wounding, despite being primarily pathogen-induced genes [2]. WRKYs can regulate plant defense responses by binding to the W-box in the promoter region of defense-related genes, such as *PR* genes, through their WRKY domain [16,18,45]. These results suggest that WRKY TFs are rapidly stimulated by wounding to regulate the plant defense responses.

The second largest wound-inducible TF group in *S. tora* was the ERFs. Thirty ERF-encoding genes were upregulated by wounding (Figure 5B and Appendix A). Previous studies have demonstrated that mechanical damage rapidly induces the expression of *ERF* genes [2,46]. These proteins contain three beta-sheet strands responsible for DNA binding and an AP2/ERF domain consisting of an alpha-helix motif [47]. The ERFs were first identified through their binding to the GCC boxes in the promoter region of genes induced by ET, which is essential for the expression of many *PR* genes [37,48]. ERF TFs also respond to stress signals such as JA, SA, and fungal attacks [2,37], suggesting that these TFs are involved in plant defense responses.

Several genes encoding MYB and bHLH TFs were induced by wounding (Figure 5C,D and Appendix A). The bHLH group is characterized as TFs containing a basic helix-loop-helix (bHLH) domain [47]. These TFs bind to the G-box sequence in the promoter region of their target genes and are involved in anthocyanin accumulation and resistance to pathogens [4,49]. The MYB TF family comprises proteins containing an alpha-helix repeat structure, which are categorized into R1, R2R3, R3, and R4 MYB proteins according to the number of repeats they possess. The R2R3 MYB family is specific to plants and responds to biotic and abiotic stresses [47]. They regulate many biosynthesis pathways in plants, including tryptophan biosynthesis, glucosinolate biosynthesis, and flavonoid biosynthesis [2,49,50].

Eighteen NAC-encoding genes were identified as DEGs after wounding (Figure 5E and Appendix A). NAC TFs are defined as proteins containing the NAC domain, which is derived from NAM (no apical meristem), ATAFs (Arabidopsis transcription factors), and CUC2 (cup-shaped cotyledon) [17,47]. They can regulate the expression of *PR* genes and are involved in plant disease resistance against to viruses and fungi [17,49]. These results suggest that wounding can induce the expression of many *TF* genes involved in the plant defense responses. In Arabidopsis, approximately 20% of all wound-induced genes encode proteins involved in the regulation of transcription and stress signaling. Among them, the ERF, WRKY, bHLH and MYB genes are the main TFs induced by wounding [2].

### 2.4. Accumulation of Transcripts Encoding PR Proteins following Wounding

We investigated expression levels of the *PR* genes following wounding in the *S*. *tora* leaves. The genes encoding PR-1, CHT, TLP, cysteine PI (CPI), PR-10, and PDF were induced by wounding (Figure 6A and Appendix A). The induction of *STO02G053750*, encoding PR-1, was identified in T6. Seven CHT-encoding genes were induced by wounding; among them, the expression of *STO09G312570*, *STO12G374780*, and *STO08G235260* was induced in T1 and peaked at T3, while *STO01G005390*, *STO01G005380*, and *STO06G201890* were induced in T3 and peaked at T24. Two *TLP1b* genes, *STO08G241320* and *STO13G435030*, were induced in T6 and peaked at T24. Six *PR-10* genes, *STO10G345160*, *STO10G345190*, *STO10G345200*, *STO03G062560*, *STO10G345180*, and *STO10G345220*, showed very similar expression patterns, being induced in T1 and peaking at T3. The expression of the CPI-encoding gene *STO05G159460* and the PDF-encoding gene *STO11G363050* was induced in T3 and peaked at T6 (Figure 6A). Although the peak expression of these PR-encoding genes occurred at different timepoints, most were induced in T3 and maintained significantly higher expression levels until T24 than were observed at T0 (Appendix A).

The accumulation of PR proteins has been reported to protect against infection by pathogens and multiple other stresses [18]. CHTs belong to groups PR-3 and PR-4, which are mainly induced by pathogenic fungi and bacteria [18,51,52]. CPI belongs to the PR-6 group. PIs accumulate systemically within 2 h of an insect attack or wounding, making them useful model genes that can explain the systemic defense mechanism in tomato (*Solanum lycopersicum*) [4]. PDF, a member of the PR-12 group, is a well-known antifungal protein in plants, the production of which is induced by stresses such as pathogenic fungi and wounding [52]. Members of the PR-10 group are known as ribonuclease-like proteins, which are upregulated by multiple abiotic stresses and exhibit broad antifungal activity against various species [52]. The accumulation of PR proteins is therefore critical to the immune response in plants [18]. Induction of the *PR* genes observed in the present study is clear evidence that wounding can induce the plant defense responses.

### 2.5. Regulation of the Flavonoid Biosynthesis Pathway by Wounding and the Accumulation of the Flavonols Kaempferol and Quercetin

All genes involved in flavonoid biosynthesis were induced by wounding in *S*. *tora* leaves (Figure 6B,C and Appendix A). Based on their expression patterns, two PAL1-encoding genes (*STO13G435570* and *STO08G240820*), a C4H-encoding gene (*RSGO05G161760*), a 4CL-encoding gene (*STO07G231630*), nine CHS-encoding genes (*STO03G063050*, *STO03G054970*, *STO03G058210*, *STO03G058230*, *STO03G058250*, *STO03G058240*, *STO08G243540*, *STO03G054960*, and *STO03G054930*), a CHI-encoding gene (*STO05G132000*), a F3H-encoding gene (*STO06G201230*), and a FLS-encoding gene (*STO09G308580*) were activated at similar timepoints after wounding. Most of these genes were induced in T1 and showed a peak expression level at T3 (Figure 6C and Appendix A). Since the expression of the FLS-encoding gene *STO09G308580* was strongly induced (Figure 6C), we performed a quantitative analysis of two well-known flavonols, kaempferol and quercetin. A significant amount of kaempferol and quercetin had accumulated at T3 (Figure 6D). The induction of flavonoid biosynthesis genes by wounding has been well documented in previous studies [2,22,32], and quercetin and kaempferol are known to possess numerous antioxidant, antimicrobial, and antifungal functions [23,53]. Phytochemical studies of *S*. *tora* revealed that flavonoid compounds, including quercetin and kaempferol, are the main components in leaves [54].

Differential expression patterns were found between the genes encoding PAL1, CHI, and CHS. The expression of the PAL1-encoding gene *STO06G182100*, the CHI-encoding gene *STO10G323650*, and seven CHS-encoding genes (*STO07G228230*, *STO07G228240*, *STO09G307690*, *SOT09G307640*, *STO07G228270*, *STO07G228320*, and *STO07G228300*) were most strongly induced in T3 and peaked at T6 (Figure 6B and Appendix A). These genes may be responsible for other pathways in addition to flavonoid biosynthesis. PAL catalyzed the deamination of phenylalanine to produce cinnamic acid, which is the first step in flavonoid biosynthesis, but it is also involved in the SA biosynthesis pathway [42,55]. CHIs are key enzymes that catalyze the intramolecular isomerization of naringenin chalcone in the flavonoid biosynthesis pathway. Among them, type II CHI is involved in the biosynthesis of isoliquiritigenin, a legume-specific metabolite [56]. CHS catalyzes the conversion of naringenin chalcone from *p*-coumaroyl CoA. A recent study demonstrated that the CHS-like gene family has been specifically expanded in *S*. *tora* to play a role in anthraquinone biosynthesis [27]. Among the seven CHS-encoding genes induced in T3, five genes (*STO07G228230*, *STO07G228240*, *STO07G228320*, *STO07G228300*, and *STO07G228270*) were annotated as CHS-like proteins (Appendix A) [27].

### 2.6. Functional Network Prediction of Wound-Inducible Genes Based on a Co-Expression Analysis

The detection of genes with similar expression patterns under various experimental conditions may imply their functional association and shared regulation [57]. In this study, 155 genes that met strict filtering thresholds (FPKM >10 and FC >3) were used for co-expression analysis. These genes were divided into four clusters based on the association of their gene expression patterns (Figure 7 and Appendix A). The genes included in cluster I are characterized by a strong induction in T1 after wounding followed by a marked decrease in expression (Figure 7A,B). The genes involved in the jasmonate biosynthesis pathway were identified as the main components of cluster I. Genes encoding bHLH, ERF, MYB, WRKY, and NAC TFs also belonged to this group (Appendix A). The bHLHs and MYBs are major regulators of jasmonate signaling, and they can directly regulate expression of the JA-Ile responsive genes [4,5,7,16]. In particular, the bHLH MYC2 is a key regulator of the transcription of JA-Ile-responsive genes [5]. Despite this, the *STO11G362400* gene annotated as *MYC2* did not show a distinct induction in this study (Appendix A). There are two possible reasons for this: first, other bHLH-encoding genes have been shown to have the potential to replace *MYC2* function [4,5,7], and second, it may not be induced owing to a lack of JA-Ile activity. The JAR1-encoding gene *STO06G173150* was not a significant DEG, but two genes encoding CYP94C1 were identified in cluster I (Figure 3B). Six JAZ-encoding genes were also included in this group (Appendix A). Like the bHLH TFs, the R2R3-type MYB TFs show a significant response to jasmonate signaling [5,7,49]. MYB21, MYB24, and MYB54 can interact with the JAZ protein through the R2R3 domain [7]. Some WRKYs are known to enhance jasmonate levels by regulating expression of the jasmonate biosynthesis genes *LOX*, *AOC*, *AOS*, and *OPR* in *Nicotiana attenuata* [5,49]. *ERF109* responds directly to jasmonate signals in Arabidopsis [58]. Arabidopsis NAC019 and NAC055 are involved in the regulation of jasmonate-mediated genes in Arabidopsis and rice [17].

Most genes belonging to cluster II were also induced in T1 after wounding, but their peak activities were identified in T3 (Figure 7A,C). Genes involved in ET and flavonoid biosynthesis, as well as genes encoding PR-10 proteins, were included in cluster II, along with MYB, ERF, NAC, bHLH, and WRKY TFs (Appendix A). ET is expected to be involved in the expression of genes belonging to cluster II, alongside the jasmonates, because the ET pathway interacts with the jasmonate pathway to co-regulate the expression of defense-related genes [2,37]. ERF TFs are important regulators of the ET signaling pathway and can be synergistically induced by JA [59]. The ORCAs respond to jasmonate signals and regulate crosstalk between jasmonate and ET signals, as well as being involved in positive jasmonate feedback [5,49]. ERF1, ORA59, ERF5, and ERF6 are involved in the jasmonate-/ET-mediated defense responses [7]. Flavonoid biosynthesis is known to be regulated by jasmonates and ET signaling [21,60]. Study of the Arabidopsis *chi* mutant, which fails to accumulate flavonols, revealed a transient negative feedback of flavonoids on jasmonate accumulation induced by wounding [61]. MYB TFs are key regulators of the flavonoid biosynthesis pathway [25,61]. MYB12 is involved in flavonol biosynthesis by regulating the expression of *CHS*, *CHI*, *F3H*, and *FLS* [25]. Tobacco (*Nicotiana tabacum*) MYB2 regulates the expression of *PAL2* by binding directly to its promoter region [62]. Plants overexpressing *MYB75* reportedly have altered kaempferol-3, 7-dirhamnoside levels after caterpillar or aphid feeding [61]. MYB9 and MYB11 regulate anthocyanin biosynthesis in apple (*Malus domestica*) [63]. Jasmonates and ET have been reported to induce the expression of *PR-10* genes, and WRKY TFs are required for their expression [64,65]. Parsley (*Petroselinum crispum*) *PR1-1*, *PR1-2*, and *PR1-3*, belonging to the PR-10 family, are stimulated by WRKY1–5 [45]. WRKYb binds directly to CaPR10 in pepper (*Capsicum annuum*), which leads to a positive defense response [45]. The overexpression of rice *WRKY30* has been reported to increase resistance to fungi by activating the expression of *PR-10* genes [66]. In addition, the GCC boxes and MRE sequences identified in the promoter regions of the *PR-10* genes suggest that ERF and MYB TFs are involved in their regulation [45].

ET biosynthesis genes; *PR* genes encoding PI, PDF, and CHT; and genes encoding bHLH, MYB, NAC, and WRKY TFs were members of cluster III (Appendix A). Most of these genes were induced in T3 after wounding and peaked at T6 (Figure 7A,D). ET regulates these *PR* genes [2,52,67], and a relatively late ET signal may be required to maintain their expression. In addition, *CHT*, *CPI*, and *PDF* are known as marker genes of JA signaling [18,52], and recognition of the ET signal with jasmonates is essential for the full wound-induced expression of the *PDF* and *PI* genes [3,38,68]. The presence of W-boxes in the promoters many *PR* genes suggests that WRKY TFs are involved in their regulation [18]. WRKY33 has been identified as a positive regulator of *PDF* and *CHT* expression in responses to jasmonates and ET [69]. *WRKY* or *NAC* overexpression led to the upregulation of the *PDF* gene in tobacco [70].

Genes belonging to cluster IV showed strong expression at T24 after wounding (Figure 7A,E). Genes encoding PR-1, CHT, and TLP were included in cluster IV, along with TF-encoding genes such as *ERF*s, *MYB*s, and *WRKY*s (Appendix A). As the expression of the two late ET biosynthesis genes remained significantly upregulated at T24 after wounding, ET is expected to be an important signal regulating the expression of genes in cluster IV. Although genes encoding PR-1 and TLP1b can be induced by ET [71,72], they are generally marker genes for SA signaling [38,52]. These results support the potential occurrence of SA biosynthesis following wounding in *S*. *tora* leaves (Figure 4). In addition, *STO02G053750*, encoding PR-1, was also annotated as an ortholog of Arabidopsis CAPE2 (Appendix A). The CAPE protein is a peptide derived from PR-1b, which is believed to be similar to systemin [72]. In tomato, *CAPE* was induced by wounding and MeJA, and a synthetic CAPE treatment induced the production of jasmonates and SA, as well as the expression of many defense-related genes [72]. This suggested that, although the jasmonate and SA biosynthesis pathways are antagonistic, Jasmonate-SA-ET can act synergistically to redirect plant defense responses [72]. These results suggest that the expression of *STO02G053750* can lead to SA biosynthesis in *S*. *tora* leaves, thereby playing a role in the plant defense response to wounding. WRKY TFs are proposed to regulate the expression of the *PR-1* genes. WRKY1 directly targets *PR1-1* in parsley, and the overexpression of *Pinella ternata WRKY293* can enhance the expression of *PR-1a* in tobacco [70,73].

In conclusion, jasmonates and ET are expected to be the main hormones induced by wounding in *S*. *tora* leaves. This damage caused the upregulation of genes encoding ERF, WRKY, NAC, MYB, and bHLH TFs, as well as those encoding PR-1, CHT, TLP, PI, PR-10, and PDF proteins. Genes involved in flavonoid biosynthesis were strongly upregulated by wounding, and an enhanced accumulation of the flavonols kaempferol and quercetin was identified in *S*. *tora* leaves. All the analyzed genes were systemically expressed in undamaged leaves. Co-expression analysis of the wound-responsive genes provided information on the functional associations between the defense-related genes, which will provide further insights into their regulation in other plants.

## 3. Materials and Methods

### 3.1. Plant Materials and Wounding Treatment

*S*. *tora* seeds were sown individually in 32 well plastic trays. After 2 weeks, the seedlings were individually transplanted into plastic pots (width 300 mm × height 270 mm) and grown in the greenhouse for 8 weeks. Wounding was achieved by removing mature leaves by cutting the petiole with scissors. The collected leaves were immediately stored in liquid nitrogen for use as the control leaves (T0) in this study. Other leaves distant from the first wound site were collected in the same manner as control leaves by cutting the petiole at 1 (T1), 3 (T3), 6 (T6), and 24 h (T24) after wounding. The samples were collected in three replicates from independent plants at each timepoint. The distance between the wound site and the undamaged leaves was set randomly.

### 3.2. RNA Preparation and Sequencing

RNA was extracted from the collected *S*. *tora* leaves using an RNeasy Plant Mini Kit (Qiagen, Hilden, Germany). The total RNA concentration was calculated using a Quant-IT RiboGreen (Thermo Fisher Scientific, Waltham, MA, USA). To assess the integrity of the total RNA, the samples were run on TapeStation RNA screentape (Agilent Technologies, Santa Clara, CA, USA). Only high-quality RNA preparations, with an RNA integrity number (RIN) greater than 7.0, were used for the RNA library construction. A library was prepared with 1 μg of total RNA for each sample using an Illumina TruSeq mRNA Sample Prep kit (Illumina, San Diego, CA, USA). The libraries were quantified using qPCR according to the qPCR Quantification Protocol Guide (KAPA Library Quantification kits for Illumina Sequencing platforms), and qualified using the TapeStation D1000 ScreenTape (Agilent Technologies, Santa Clara, CA, USA). Indexed libraries were then sequenced using a HiSeq4000 platform (Illumina) by Macrogen Incorporated (Seoul, Korea).

### 3.3. Processing of RNA Sequencing Data

The raw reads from the sequencer were preprocessed to remove low-quality and adapter sequences before the analysis. The processed reads were aligned using HISAT v. 2.1.0 [74]. Transcript assembly and abundance estimation were achieved using StringTie v. 1.3.4d [75,76]. The protein coding region in the *S*. *tora* genome was used as a reference sequence for mapping [27]. The FPKM values were calculated using the number of RNA-seq reads mapped onto the reference sequences.

### 3.4. Statistical Analysis of the DEGs

The relative abundances of the genes were measured in Read Counts using StringTie. Statistical analysis was performed to identify the DEGs using estimates of the abundances for each gene in the samples. The statistical significance of the DEGs was determined using nbinomWaldTest in DESeq2 [77]. Genes with an FC over 2 with an adjusted *p*-value less than 0.05 when compared with the control sample were considered significant DEGs. The false discovery rate (FDR) was controlled for by adjusting the *p*-value using the Benjamini–Hochberg algorithm. For the DEG set, a hierarchical clustering analysis was performed using complete linkage and Euclidean distance as a measure of similarity. Expression levels in the heatmap were measured after transformation into a Z-score by R package pheatmap [78].

### 3.5. GO and KEGG Enrichment Analysis, and Co-Expression Analysis

GO enrichment analysis was performed for the DEGs using Fisher’s exact test, after which the enriched GO terms were classified into over- and underrepresented terms using BLAST2GO (v. 5.2.5; Available online: https://www.blast2go.com/ (accessed on 1 July 2020)). GO terms with a gene-rich factor > 2 at *p*-value < 0.01 and FDR < 0.01 were further selected as enriched GO terms and visualized using the ggplot function provided by the R package ggplot2. The gene-rich factor was calculated as the ratio of the DEGs assigned to the GO term/all genes assigned to that GO term.

KEGG pathway enrichment analysis was carried out using a modified Fisher’s exact test performed with the fisher.test function in R. KEGG pathways with a gene-rich factor > 2 at *p*-value < 0.01 and FDR < 0.01 were further selected as enriched KEGG pathways and visualized using the ggplot function provided by the R package ggplot2. The gene-rich factor was calculated as the ratio of the DEGs assigned to the KEGG pathway/all genes assigned to that KEGG pathway.

A co-expression analysis was performed using R studio. DEGs with FPKM > 10 and FC > 3 were selected for the analysis. The heatmap of the selected DEGs was analyzed using the Euclidean method in the pheatmap package based on the Z-score, and the line graphs were visualized using the ggplot2 package with the log_2_ FPKM of the selected DEGs.

### 3.6. Quercetin and Kaempferol Content Analysis

For quantitative analysis of quercetin and kaempferol, leaf powder for each stage was extracted by vortexing for 3 h in a 1:1 mix of methanol:chloromethane. The extract was filtered using a 0.22 μm Whatman syringe filter (Whatman GmbH, Dassel, Germany), then concentrated under reduced pressure. The sample was diluted 20 times using methanol, then analyzed using an Agilent 1100 series high-performance liquid chromatography (HPLC) system (Agilent Technologies). The separation was performed using a Shim-pack GIS-ODS column (4.6 × 250 mm, 5 μm), and the data were collected using Agilent ChemStation software (Agilent Technologies, Santa Clara, CA, USA). The mobile phase consisted of solution A (HPLC-grade water and 0.075% trifluoroacetic acid) and solution B (HPLC-grade acetonitrile). The flow rate was 1.0 mL/min, and the oven temperature was maintained at 25 °C. The extract was analyzed by increasing the ratio of solution B to solution A from 10 to 50% (0–8 min), then 70% (14 min), then 90% (15 min), then 10% (20 min), and holding at 10% for 20–22 min. The substrates and extract were confirmed using UV detection at 254 nm.

## Figures and Tables

**Figure 1 ijms-22-10073-f001:**
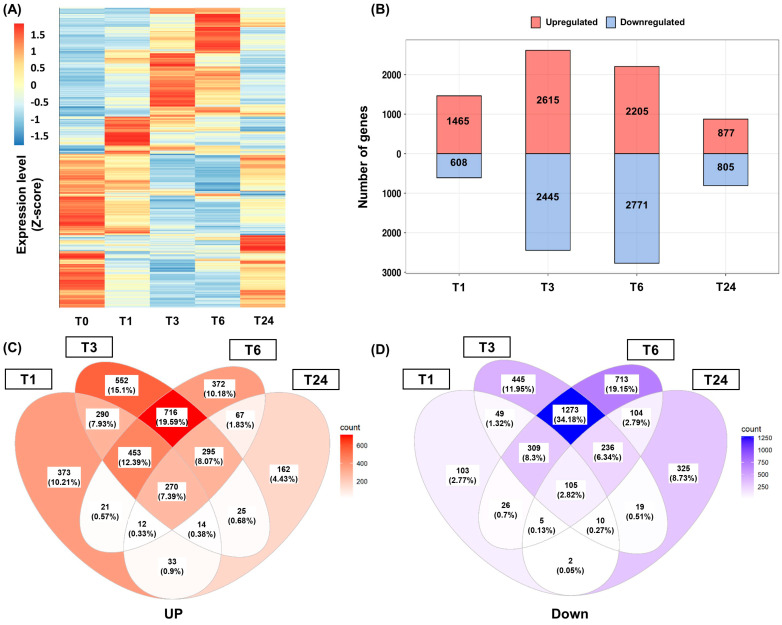
Analysis of the differentially expressed genes (DEGs) following wounding in *S*. *tora* leaves. (**A**) Total DEGs analysis using a heatmap. The fragments per kilobase of transcript per million mapped reads (FPKM) value for each individual gene was normalized using the Z-score, and a heatmap was generated by setting the clustering method ‘complete’ using the pheatmap package in R. (**B**) Number of up and downregulated DEGs following wounding at each timepoint. (**C**,**D**) Venn diagrams of the up (left) and downregulated DEGs (right) following wounding. T0, T1, T3, T6, and T24 are the timepoints at 0, 1, 3, 6, and 24 h after wounding, respectively.

**Figure 2 ijms-22-10073-f002:**
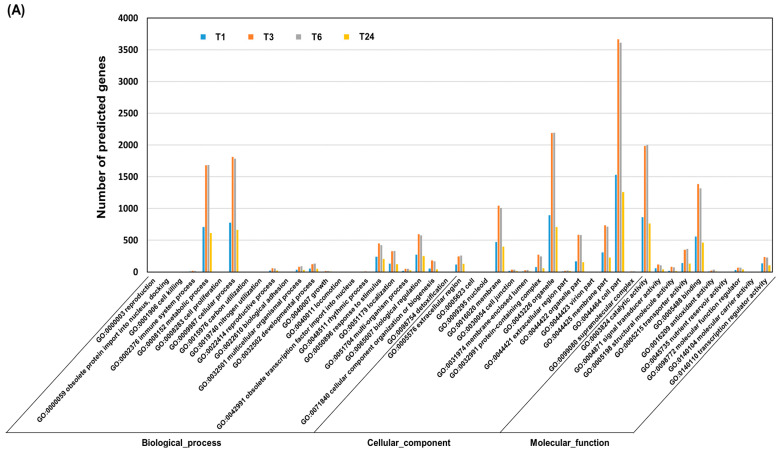
Gene ontology (GO) and Kyoto Encyclopedia of Genes and Genomes (KEGG) pathway enrichment analysis of the differentially expressed genes (DEGs) following wounding of *S*. *tora* leaves. (**A**) Histogram of the GO classification of the DEGs. The results are summarized in three main categories: biological process, cellular component, and molecular function. (**B**,**C**) Dot plot of the GO enrichment of the DEGs. The color of each circle represents the *p*-value of the GO terms associated with the DEGs at that timepoint compared with T0. The size of the circle indicates the gene-rich factor, which was defined as the ratio of the DEGs assigned to the GO term/all genes assigned to that GO term. (**D**,**E**) Dot plot of KEGG pathway enrichment for the DEGs. The color of the circle represents the *p*-values of the KEGG pathway terms associated with the DEGs at that timepoint compared with T0. The size of the circle represents the gene-rich factor, which was defined as the ratio of the DEGs assigned to the KEGG pathway term/all genes assigned to that KEGG pathway term. The dot plots were generated using the ggplot function in the R package ggplot2.

**Figure 3 ijms-22-10073-f003:**
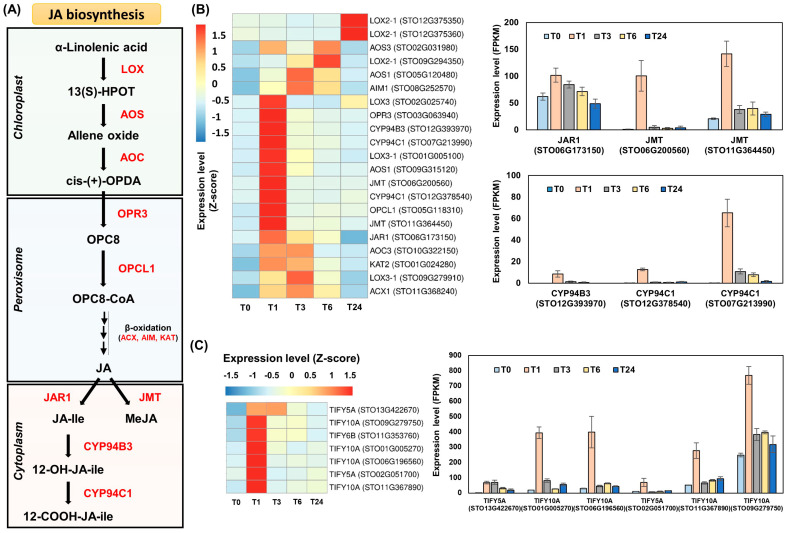
Jasmonate biosynthesis genes are induced by wounding in *S*. *tora* leaves. (**A**) Jasmonate biosynthesis pathway in plants. 13(S)-HPOT, alpha-linolenic acid 13-hydroperoxide; OPDA, oxo-phytodienoic acid; OPC8, 3-oxo-2-(2′(Z)-pentenyl)-cyclopentane-1 octanoic acid; OPC8-CoA, 3-oxo-2-(2′(Z)-pentenyl)-cyclopentane-1 octanoic acid coenzyme A; JA, jasmonic acid; JA-Ile, jasmonoyl-isoleucine; MeJA, methyl jasmonate; LOX, lipoxygenase; AOS, allene oxide synthase; AOC, allene oxide cyclase; OPR3, OPDA reductase; OPCL1, 4-coumarate-CoA ligase; ACX, acyl-coenzyme A oxidase; AIM, fatty acid beta-oxidation multifunctional protein; KAT, 3-ketoacyl-CoA thiolase; JAR1, jasmonate resistant 1; JMT, JA carboxyl methyltransferase; CYP, cytochrome P450. (**B**) Heatmap and bar graphs of genes involved in the jasmonate biosynthesis pathway. Genes responsible for JA metabolism are shown in bar graphs (right). (**C**) Expression analysis of genes encoding TIFY inhibitors. TIFY-encoding genes filtered by a fold change (FC) > 3 are shown in a bar graph (right). The fragments per kilobase of transcript per million mapped reads (FPKM) values of the individual genes were normalized to the Z-score. The heatmap was visualized using the pheatmap package in R. Statistical analysis was performed on the expression levels of each gene expressed at the different times compared with T0 (Appendix A).

**Figure 4 ijms-22-10073-f004:**
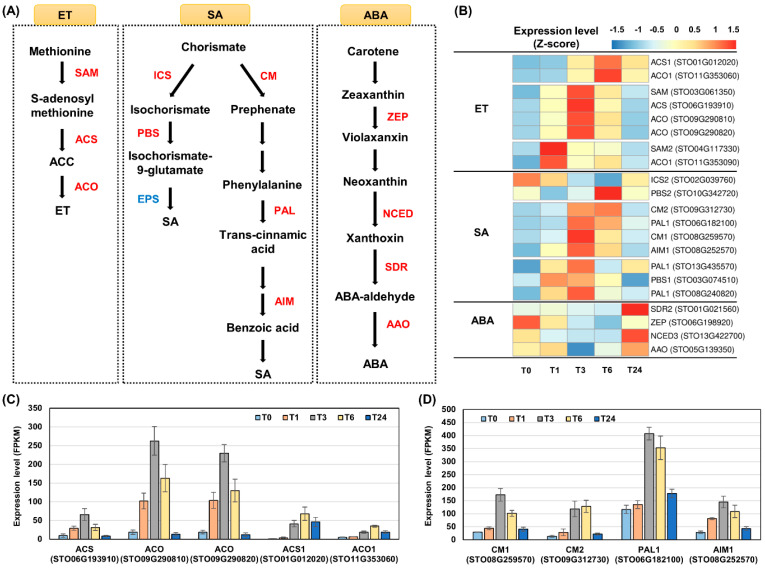
Expression analysis of genes involved in ethylene (ET), salicylic acid (SA), and abscisic acid (ABA) biosynthesis induced by wounding in *S*. *tora* leaves. (**A**) ET, SA, and ABA biosynthesis pathways in plants. (**B**) Heatmap analysis of genes involved in ET, SA, and ABA biosynthesis. ACC, 1-aminocyclopropane-1-carboxylate; SAM, S-adenosylmethionine synthase; ACS, 1-aminocyclopropane-1-carboxylate synthase; ACO, 1-aminocyclopropane-1-carboxylate oxidase; ICS, isochorismate synthase; PBS, PphB SUSCEPTIBLE; EPS, ENHANCED PSEUDOMONAS SUSCEPTIBILITY; CM, chorismate mutase; PAL, phenylalanine ammonia-lyase; AIM, fatty acid beta-oxidation multifunctional protein; ZEP, zeaxanthin epoxidase; NCED, 9-cis-epoxycarotenoid dioxygenase; SDR, short-chain dehydrogenase/reductase; AAO, ABA aldehyde oxidase. The gene encoding the SA biosynthesis enzyme EPS (blue color) was not identified in this study. The fragments per kilobase of transcript per million mapped reads (FPKM) values of the individual genes were normalized to the Z-score, and the heatmap was visualized using the pheatmap package in R. (**C**) Expression of the ET biosynthesis genes. (**D**) Bar graph analysis of genes involved in second SA biosynthesis pathway. Statistical analysis was performed on the expression levels of each gene expressed at the different times compared with T0 (Appendix A).

**Figure 5 ijms-22-10073-f005:**
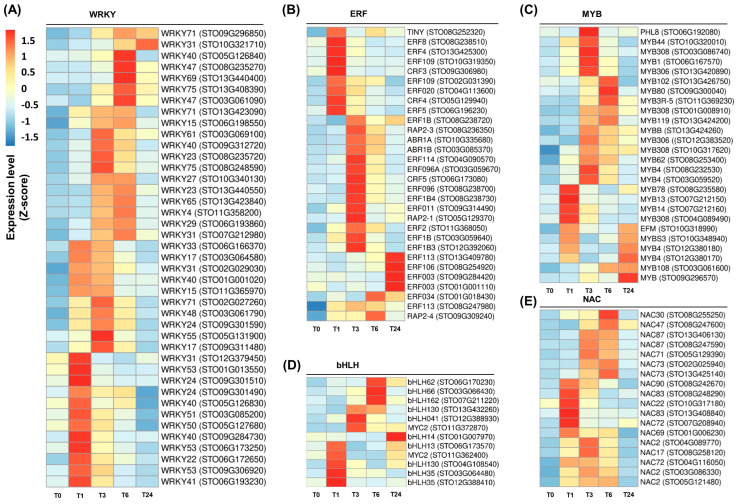
Induction of transcription factor (TF)-encoding genes following wounding in *S*. *tora* leaves. (**A**–**E**) Heatmap analysis of genes encoding WRKY (**A**), ERF (**B**), MYB (**C**), bHLH (**D**), and NAC (**E**) TFs. The fragments per kilobase of transcript per million mapped reads (FPKM) value for each individual gene was normalized to the Z-score. The heatmap was visualized using the pheatmap package in R. Statistical analysis was performed on the expression levels of each gene expressed at the different times compared with T0 (Appendix A).

**Figure 6 ijms-22-10073-f006:**
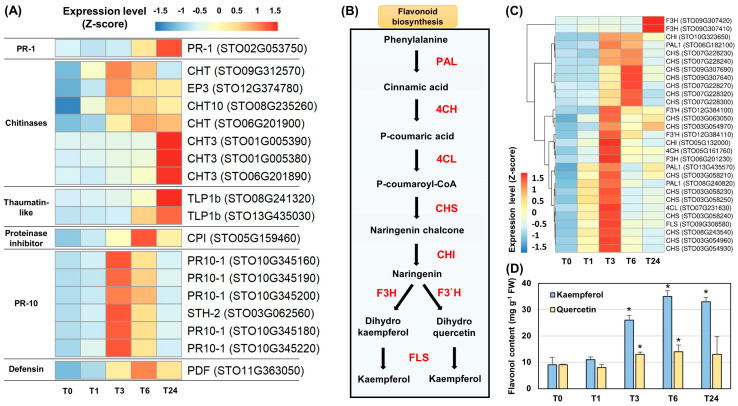
Expression analysis of genes encoding pathogenesis-related (PR) proteins and those involved in flavonoid biosynthesis induced following wounding in *S*. *tora* leaves. (**A**) Heatmap analysis of genes encoding PR proteins. (**B**) Flavonoid biosynthesis pathway in plants. PAL, phenylalanine ammonia-lyase; C4H, cinnamate 4-hydroxylase; 4CL, 4-coumaroyl-CoA ligase; CHS, chalcone synthase; CHI, chalcone isomerase; F3H, flavonone 3-hydroxylase; F3′H, flavonoid 3′-monooxygenease; FLS, flavonol synthase. (**C**) Heatmap analysis of genes involved in flavonoid biosynthesis. The fragments per kilobase of transcript per million mapped reads (FPKM) values for each individual gene were normalized to the Z-score. The heatmap was visualized using the pheatmap package in R. A statistical analysis was performed on the expression levels of each gene expressed at the different times compared with T0 (Appendix A). (**D**) Accumulation of kaempferol and quercetin. Significant differences compared with T0 are indicated as * *p*-value < 0.05 (Student’s *t* test).

**Figure 7 ijms-22-10073-f007:**
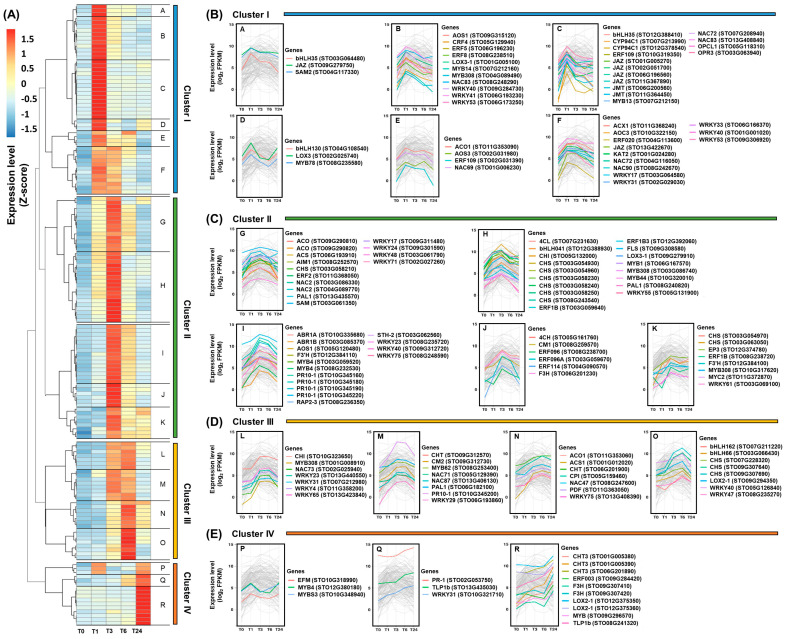
Co-expression analysis of the differentially expressed genes (DEGs) induced by wounding in the *S*. *tora* leaves based on the expression patterns. (**A**) Heatmap analysis of selected DEGs. The fragments per kilobase of transcript per million mapped reads (FPKM) values for each individual gene were normalized to the Z-score. The heatmap was visualized using the pheatmap package in R. Statistical analysis was performed on the expression levels of each gene expressed at the different times compared with T0 (Appendix A). (**B**–**E**) Analysis of the co-expression of selected DEGs using a line graph. The expression levels were calculated using the log_2_ scale of each gene FPKM. Genes with a fold change below three and a FPKM below 10 were excluded from the co-expression analysis.

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
