# Peer review of "Systemic Expression of Genes Involved in the Plant Defense Response Induced by Wounding in *Senna tora"

_ijms, 2021, doi:10.3390/ijms221810073_

Round 1
Reviewer 1 Report
Manuscript ID: ijms-1360062
Title: Systemic Expression of Genes Involved in the Plant Defense Response Induced by Wounding in Senna tora
Authors: Kang Ji-Nam, et al.
In this manuscript, Kang et authors investigate through RNA sequencing and transcriptomic analysis the molecular responses of Senna to wounding. Genes and pathways induced by the wound damage are analyzed, and roles of individual hormone-related response pathways are analyzed and discussed.
The work is scientifically sound, the experimental design is clear and quite straightforward. Methods and results are clearly described. Results are thoroughly exposed and commented.
The major drawback of the Kang and colleagues' manuscript is the lack of any analysis and comment about downregulated genes. Although they demonstrated that, particularly at T3 to T24, the number of dowregulated genes equalled or exceeded that of upregulated genes, the whole analysis presented in this work focuses on induced genes. On one side this is coherent with the object of the study and the title itself, that concentrates on plant defense response pathways that are transcriptionally induced after wounding. On the other side, some defense pathways and other important metabolic functions and biological processes may be repressed as well. Therefore, I strongly recommend the authors to complete their significant effort by dedicating at least an independent new paragraph to genes, pathways and processes that are transcriptionally repressed in their case study.
Other minor changes are suggested as follows:
Line 89: There is no mention to how "Expression level (Z-score)" are calculated in Materials and methods. A brief explanation, a formula or a citation may help.
- 90: Do FC values show up in any particular order in the Fig. 1A graph? Are they obtained by hierarchical clustering? Please give some more details in the caption.
- 100: See the comment above. Why downregulated genes where wholly excluded by any further analysis?
- 141: "The additional wound treatment": from Material and methods it seems that there was only one wound treatment, after which leaf samples were collected at different time points. Please explain in clear, understandable and unequivocal form the procedure that was followed.
- 429: "The distance was set randomly": indicate a range of distances from the original wounding site
Author Response
Thank you for your comments.
We have carefully revised the manuscript based on the comments and are responding to each comment fully below.
Responses are submitted as word file.
Please see the attachment.

Reviewer 2 Report
The submitted manuscript entitled " Systemic Expression of Genes Involved in the Plant Defense Response Induced by Wounding in Senna tora" submitted by Kang et al., is a research article with up-to-date literature information in the Introduction part. The research article describes mechanical wounding can lead to a systemic defense response by the induction of genes involved in flavonoid biosynthesis and the accumulation of kaempferol and quercetin in S. tora leaves. This is the novel study work and the results are supporting the hypothesis. Moreover, the Conclusion is supported by the results and discussion. The research manuscript as written suits the requirements and the standard of the IJMS journal.
Abstract: Abstract is well written and depicts the whole story of the manuscript.
Introduction-It is well written with up-to-date scientific literature with following the logic.
Results and discussion-Authors wrote descriptive results and discussed them with scientific evidence.
Material and methods-Also, well written and easily understandable.
Hence, I would like to recommend this paper for further process.
Author Response
Thanks for your comment.
Round 2
Reviewer 1 Report
The authors presented a revised manuscript that responds to reviewers' suggestion, and has been significantly improved. I propose the manuscript for publication on this journal, and only suggest some very minor text editing, as below:
L. 113: change "invovled" to "involved"
L. 177: "which important for plant growth..." please reword for correct English
Fig. 2 B and C: Consistently indicate GO codes in both panels
Author Response
Thank you for your detailed review.
We revised the word "invovled" on line 113 to be "involved".
We revised the sentence on line 117 by adding the verb (is).
We added the GO code in Figure 2B.